# Structural Evidence of Programmed Cell Death Induction by Tungsten in Root Tip Cells of *Pisum sativum*

**DOI:** 10.3390/plants8030062

**Published:** 2019-03-11

**Authors:** Ioannis-Dimosthenis S. Adamakis, Eleftherios P. Eleftheriou

**Affiliations:** 1Department of Botany, School of Biology, National and Kapodistrian University of Athens, 157 84 Athens, Greece; iadamaki@biol.uoa.gr; 2Department of Botany, School of Biology, Aristotle University of Thessaloniki, 541 24 Thessaloniki, Greece

**Keywords:** endoplasmic reticulum, *Pisum sativum*, programmed cells death, reactive oxygen species, tungsten, ultrastructural malformations, vacuolar collapse

## Abstract

Previous studies have shown that excess tungsten (W), a rare heavy metal, is toxic to plant cells and may induce a kind of programmed cell death (PCD). In the present study we used transmission electron microscopy (TEM) and confocal laser scanning microscopy (CLSM) to investigate the subcellular malformations caused by W, supplied as 200 mg/L sodium tungstate (Na_2_WO_4_) for 12 or 24 h, in root tip cells of *Pisum sativum* (pea), The objective was to provide additional evidence in support of the notion of PCD induction and the presumed involvement of reactive oxygen species (ROS). It is shown ultrastructurally that W inhibited seedling growth, deranged root tip morphology, induced the collapse and deformation of vacuoles, degraded Golgi bodies, increased the incidence of multivesicular and multilamellar bodies, and caused the detachment of the plasma membrane from the cell walls. Plastids and mitochondria were also affected. By TEM, the endoplasmic reticulum appeared in aggregations of straight, curved or concentric cisternae, frequently enclosing cytoplasmic organelles, while by CLSM it appeared in bright ring-like aggregations and was severely disrupted in mitotic cells. However, no evidence of ROS increase was obtained. Overall, these findings support the view of a W-induced vacuolar destructive PCD without ROS enhancement.

## 1. Introduction

The earth’s lithosphere is rich in many metals and metalloids, which have been employed by mankind since the dawn of civilization for making innumerous kinds of tools, machinery, weapons, homes and appliances. Due to these extensive anthropogenic activities, which have exponentially increased over the last three centuries, many heavy metals and their compounds were and are accumulated locally at high concentrations. Many of them, if not all, were proved to cause harmful effects to the environment, living organisms and humans to varying degrees, depending on the kind of metal, concentration and environmental conditions [1,2,3].

One of the heavy metals that has attracted scientific interest only in recent decades—presumably due to its rarity in nature—for its potential toxicity is tungsten (W), a scarce transition metal [4,5]. Elevated concentrations of W at harmful levels to biota may have a natural or anthropogenic origin [6,7]. Since W displays some unique physical and chemical properties (high density, high melting point, high conductivity, hardness, resistance to oxygen, acids and alkalis), it is suitable for a broad spectrum of household, industrial, scientific and military applications [4,5]. Moreover, the use of some phosphate fertilizers that may contain W in agriculture [8] and the introduction of W in military and war practices as an alternative to lead and depleted uranium [9,10] have increased W concentrations in surface soil and human accessibility [11], raising concerns about its potential adverse public health effects [5,7,12]. Extensive medical research has produced voluminous information concluding that W may cause serious health effects to humans and animals, especially provoked after the observed childhood leukemia cluster since the late 1990s in some areas of the USA neighboring W factories or military facilities [13,14,15]. Although no direct relationship between the leukemia cluster and W exposure was shown unequivocally, extensive research was boosted on the effects of W on microbes, animals and humans [16,17,18].

On the other hand, far fewer studies have been conducted on the potential toxicity of W to plants, most of which are related to molybdeno-enzyme (Mo-enzyme) research where W is used as an inhibitor of Mo-enzymes [19,20]. Further studies have shown that W not only affects Mo-enzymes but additionally causes many other aberrations [21,22]. In particular, W was shown to induce a series of adverse effects ranging from impediment of seedling growth [23] to the disorganization of microtubules (MTs) and actin microfilaments (AFs) [24,25,26]. Moreover, it was found that W impaired the distribution of auxin transporters [27] and induced a kind of programmed cell death (PCD) [28,29]. PCD is an integral part of plant development and defense. Developmental PCD is an essential process during the differentiation of some tissues such as xylem or organs, including roots, or the morphogenesis of perforated or highly indented leaves [30,31]. PCD may also be triggered by both environmental abiotic and biotic stresses and is frequently accompanied by the increased generation of reactive oxygen species (ROS) that seriously reduce plant productivity [32,33,34]. Root tip cells of *Pisum sativum* exposed to W displayed a shrinkage of protoplasts, chromatin condensation at the nucleoplasm periphery, increased absorption of the stain Evans blue and enhanced special gene expression, responses which are considered to be a kind of PCD [28,29]. However, the ultrastructural effects of W toxicity have not been explored satisfactorily in relation to PCD induction, neither is it clear if they are accompanied by ROS production.

Therefore, the aim of the present study was to search for additional evidence in support of the notion of PCD, including the presumed enhancement of ROS production under W stress. For this reason, we investigated the ultrastructural malformations caused by W to cytoplasmic components besides the nucleus [28] in root epidermal and cortex cells in the meristematic zone of *P. sativum*. Since previous evidence involved endoplasmic reticulum (ER) stress-unfolded protein response in W-induced PCD [28], we also studied the distribution and conformation of ER under W stress with transmission electron microscopy (TEM) and confocal laser scanning microscopy (CLSM). 

## 2. Results

### 2.1. Macroscopic and Microscopic Effects

As expected, the preliminary experiments showed that untreated seedlings developed long primary roots and numerous lateral ones, all having a clear white color (Figure 1A). On the other hand, the W-treated seedlings displayed a concentration- and time-dependent growth decline of both shoots and roots. The shoots remained short and the leaves were folded and small, but kept their original green color. The primary roots suspended lengthening and the lateral roots did not emerge or they did not lengthen if they had already been formed before treatment. Roots of the W-treated seedlings showed a brownish color as compared to the controls (Figure 1A). Prolonged treatments with the higher concentration (500 mg/L W) had severe effects, so the lower concentration of 200 mg/L W and shorter exposures (12 or 24 h) were adopted for the main experiments. 

Central longitudinal sections of control roots displayed the typical apical organization of dicotyledons, with a straight pointed root cap, and meristematic, transition and elongation zones with neatly arranged cells in vertical files (Figure 1B). In W-treated seedlings, the root tip was deformed, the root cap was diminished and crooked, the meristematic zone was compacted and the cell files were severely disrupted (Figure 1C; cf. Figure 1B). In particular, the peripheral root cap and epidermal cells were discontinuous and disrupted. Cells at the height of the transition zone were highly vacuolated and disarranged. In the meristematic zone of the control roots, numerous mitotic phases were seen (Figure 1D), which could not be encountered in W-treated roots (Figure 1E). Instead, evident detachments of the plasma membrane from the cell wall were observed in many cells, particularly in the root epidermis but also in the external cortex cells (Figure 1E).

### 2.2. Ultrastructural Malformations

TEM revealed a plethora of ultrastructural malformations of W-treated cells compared to control cells. In the untreated seedlings, cortex cells of the meristematic root tip zone were rich in cytoplasm abounded with organelles such as plastids, mitochondria, Golgi bodies and ER (Figure 2A). The vacuoles were small, scattered throughout the cytoplasm among organelles, clearly delimited by an intact tonoplast and enclosing some sparse granulo-fibrillar material (Figure 2A). Progressively, they tended to form one or a few central vacuoles. In W-treated roots, some vacuoles retained their integrity but most presented various aberrations. Even at mild treatments (12 h), some vacuoles bore large engulfments caused by local retraction of the tonoplast, filled with granular material (Figure 2B). Occasionally, concentric multilamellar aggregations were found within the vacuoles and the engulfments (Figure 2B, and inset). Invaginations of amorphous, vesicular or membranous elements of cytoplasmic origin were also encountered (Figure 2C). Moreover, vacuoles with contracted tonoplasts that had irregular outlines, confining condensed granular material, were common (Figure 2D). The ex-vacuolar volume surrounding the contracted tonoplast was occupied by ribosome-like particles (Figure 2D), presumably resulting from cytoplasmic dilution.

At longer exposures (24 h), additional and more severe vacuolar malformations were observed. Some cells contained collapsed vacuoles with their central area occupied by irregular membranous conformations enclosing very dense amorphous material (Figure 2E). In others, the vacuoles were traversed with several invaginations and transvacuolar cytoplasmic strands surrounded by dense granular material (Figure 2F) rather than fibrillar (cf. Figure 2A). Cytoplasmic components such as mitochondria or plastids may have been located within the transvacuolar cytoplasmic strands (Figure 2G), while the vacuolar face of the tonoplast was lined by numerous electron-dense deposits (Figure 2F–H, Figure 3B). Further, vacuoles containing several smaller vacuoles, dense amorphous material, or ruptured membranes were also seen (Figure 3D; and data not shown).

ER in the untreated cells consisted of many independent cisternae scattered throughout the cytoplasm (Figure 2A). In 12-h W-treated cells, most ER occurred in bundles of roughly parallel cisternae, either straight or curved, among other organelles and vacuoles (Figure 3A). In longer treatments (24 h), ER appeared in concentric conformations of ring-shaped, ribosome-bearing cisternae, frequently enclosing cytoplasmic organelles such as plastids (Figure 3B), Golgi bodies or mitochondria (data not shown), extending in the ribosome-rich cytoplasm or near vacuoles (Figure 3B). Clusters of branched tubular elements containing dense amorphous material were occasionally encountered; they were delimited by intact or ruptured membranes and appeared to be continuous with ER-like cisternae (Figure 3C), denoting that they may be of ER origin.

The central area of many W-treated cells, that is normally occupied by a single large vacuole, bore unusual accumulations of electron-dense material with translucent enclosures of unclear origin, all encircled by many small vacuoles (Figure 3D). Higher magnifications revealed that the enclosures consisted of multivesicular clusters embedded in amorphous electron-dense material, amid a thick granular background (Figure 3E). Multivesicular bodies were also localized at the cell periphery, apparently resulting by inward bulging of the plasma membrane that created a cell wall–plasma membrane interspace filled with vesicles and an electron-dense matrix (Figure 3F). Similar structures containing numerous folded or concentric membranes with a visible tripartite elementary substructure (multilamellar bodies) were occasionally encountered in contact with the cell wall (Figure 3G).

Golgi bodies in the meristematic zone of the untreated seedlings were plentiful, consisting of 5–6 straight cisternae, with discernible *cis*–*trans* polarity, and surrounded by numerous vesicles denoting their active state (Figure 4A). Occasionally, ER cisternae, covered with ribosomes, neighbored the *cis* face of the Golgi stacks. Under W exposure, Golgi bodies were severely disturbed during both the 12- and 24-h treatments. In cortex cells, the Golgi bodies consisted of loose highly contrasting cisternae; while the *cis*–*trans* polarization was yet discernible, there were no surrounding vesicles which is indicative of functional inactivity (Figure 4B,C). No proximity with ER could be seen. In epidermal cells, the Golgi bodies were highly degenerated, consisting of short electron-dense cisternae, with obscure polarity and lacking any peripheral vesicles (see Figure 5C).

Mitochondria of the control cells displayed a normal ultrastructure and were delimited by intact bilayer membranes enclosing abundant cristae (Figure 4D). They were small in size and oval or pleiomorphic in shape, readily distinguished from plastids by their rich internal cristae. After W exposure, mitochondria displayed rounded or oval shapes and contained fewer yet discernible cristae in a less dense matrix (Figure 4E,F).

Plastids of the control cells bore rudimentary thylakoids, large starch grains and some plastoglobuli, features that distinguished them from the nearby mitochondria (Figure 2A, Figure 4G; cf. Figure 4D). Occasionally, they contained electron-dense accumulations, presumably of protein deposits (Figure 4G). They were round, oval or pleiomorphic in shape and small in size, but larger than mitochondria. After W treatment, the cortex cell plastids were spherical or oval in shape, rarely dumbbell-shaped, and were delimited by an intact bilayer envelope, few, if any, thylakoids and less dense stroma than controls (Figure 4H,I; cf. Figure 4G). In the root epidermal cells in the meristematic zone, the plastids appeared much more degraded than in the cortex cells, displaying vague stroma, no thylakoids and ruptured envelope membranes (see Figure 5C). Epidermal cells appear to be more severely affected as they come into direct contact with the stressing solution.

All cells of the W-treated roots were affected, but a closer scrutiny of the meristematic zone revealed some differences between epidermal and cortex cells. Root epidermal cells suffered most and displayed frequent detachments of the plasma membrane from the cell wall that created small or large interspaces devoid of morphological components except for some granules (Figure 5A). Fewer detachments were observed in the outer cortex cells; however, no detachments were found in the inner cells (Figure 5B), and images in accordance with these findings are shown in the light micrographs (cf. Figure 1E). Higher magnification views revealed that the cytoplasmic organelles of epidermal cells such as mitochondria, plastids and Golgi bodies were severely degenerated (Figure 5C), whereas they were less severely affected in cortex cells (Figure 5D), as already described. These discrepancies are attributed to the direct contact of epidermal cells with the tungstate solution.

### 2.3. Effects of W on ER Distribution and ROS Production

We supplementarily investigated the effects of W on ER of cortex cells by CLSM after special ER immunolabeling. In interphase cells of the untreated roots, the ER was localized throughout the cytoplasm and at the perinuclear surface (Figure 6A). The nuclear envelope was also labeled, in the context that it is a special form of ER [35,36]. No large accumulations were detected in interphase cells. In mitotic cells, ER morphology and distribution strongly reflected the well-known arrays of MTs in the untreated cells of *P. sativum* for the respective phases [37]. In preprophase and prophase cells at the equatorial plane, where a preprophase band of MTs (MT-PPB) is organized, ER formed an ER preprophase band (ER-PPB) symmetrically girdling the nucleus (Figure 6B). The signal was also localized at the nuclear surface (Figure 6B), largely coinciding with the perinuclear MT spindle organized in late prophase. At metaphase and anaphase, the ER formed a metaphase and anaphase spindle (Figure 6C,D), while it abounded in the phragmoplast margins, also occurring at the new nuclei periphery (Figure 6E) in telophase, all reflecting the respective MT arrays [37].

Root tip cells of the W-treated seedlings at interphase contained bright, unusually large ER accumulations, some of which displayed ring-like shapes (Figure 6F). The ER signal was weak and rather uniform in other cytoplasmic regions, unlike in the untreated ones (Figure 6F; cf. Figure 6A). Despite the screening of innumerous cell populations produced by squashing root tips onto the polylysine-coated coverslips, mitotic phases were scarce and almost all of them were blocked at prophase. In these images, the ER signal was weak, did not reflect the expected MT organization and only a few bright aggregations could be observed (Figure 6G–I). 

The presumed induction of ROS production by W was studied by the detection of hydrogen peroxide (H_2_O_2_) with the stain 2,7-dichlorofluorescein diacetate (DCF-DA), a specific H_2_O_2_ detector. Root tips of the untreated seedlings bore only a few cells with an intense DCF-DA signal (Figure 7A), denoting low H_2_O_2_ concentrations. In W-treated seedlings, the DCF-DA image was similar to the control (Figure 7B), suggesting that W did not increase H_2_O_2_ production. On the other hand, the positive control seedlings treated with exogenously provided H_2_O_2_ displayed the intense staining of numerous cells (Figure 7C).

The MTs in interphase cells of the untreated seedlings were typically arranged in dense, parallel, transversely aligned cortical arrays (Figure 7D). In N-acetylcysteine (NAC)-treated seedlings, MT organization was quite similar to that in the untreated ones (Figure 7E; cf. Figure 7D), denoting that the MT arrays were not affected by NAC. On the contrary, in the W-treated seedlings, the MTs were severely disintegrated (Figure 7F), as was previously reported [24,25,28]. When the NAC-treated seedlings were further exposed to NAC+W, the MTs were again severely disorganized (Figure 7G), similar to W-treated seedlings (cf. Figure 7F), showing that the destructive action of W on the MTs was not prevented by NAC, further indicating that the MTs were not affected by the production of ROS. Moreover, cell vitality after NAC treatment was not significantly deteriorated as evidenced by the similar optical density of Evans blue extract with that of the untreated seedlings (Figure 7H). However, in the W-treated and NAC+W-treated seedlings, the optical density of the extracts was statistically significantly higher than for the control, but they were similar to each other (Figure 7H), suggesting that NAC did not inhibit W action, providing additional evidence in support of the view that W did not induce ROS generation.

## 3. Discussion

Previous studies on the ultrastructural effects of W in *P. sativum* were restricted to dividing cells and the nuclei, revealing the degeneration of interphase and phragmoplast MTs, malformation of cell plates, incomplete new cell walls, swelling of nucleoli, intranuclear entrapment of cellular components [23], chromatin condensation just beneath the nuclear envelope and intranuclear compartmentalization [28], indicating that a kind of W-induced PCD took place. 

Here, we report additional ultrastructural W-related damage to the cytoplasmic organelles and especially to the vacuoles of epidermal and cortex cells in the meristematic zone that further support the notion of a PCD induction. There are different types of plant PCD, the classification of which has been a matter of extensive discussions [31,34,38,39,40,41]. Plants use vacuoles and vacuolar content for PCD in two different ways: a destructive way and a non-destructive way [38,39]. Destruction is caused by vacuolar membrane collapse, followed by the release of vacuolar hydrolytic enzymes into the cytosol, resulting in rapid and direct cell death. The non-destructive way involves fusion of the vacuolar and the plasma membranes, which allows vacuolar defense proteins to be discharged into the extracellular space where the bacteria proliferate. Intriguingly, both ways use enzymes with caspase-like activity. Cell death triggered by vacuolar collapse uses vacuolar processing enzymes (VPEs) with caspase-1-like activity and is unique to plants as it has not been seen in animals [38,39,42]. In a previous study on *P. sativum*, caspase-like proteases were shown to participate in the W-induced PCD [28]. In the present study, we have repeatedly observed collapsed vacuoles, in which the usually intact vacuolar membrane was contracted in the centre, resulting in an irregular shape, and enclosed condensed material, leaving a perivacuolar space occupied by granulofibrillar material, apparently of cytoplasmic origin (Figure 2D,E). 

We also observed the engulfment of cytoplasmic material containing multilamellar bodies, vesicles or organelles such as mitochondria inside the vacuoles (Figure 2B,C,F,G). These structures resembled those observed in cases of plant microautophagy [41]. The above notion, further supported by the deformation or degeneration of cytoplasmic organelles (Figure 3A–E and Figure 4B,C,E,F,H,I), consolidated the notion of a destructive way of PCD in W-treated cells of *P. sativum*. In conclusion, the structural evidence obtained is compatible with the view of a vacuolar destructive PCD [38,39] since a fusion of the tonoplast with the plasma membrane was never observed; similarly with an ultrastructural analysis [42], vacuolar collapse in association with the plasmolysis and formation of cytoplasmic aggregations within the cells was also observed. Additionally, vacuolar collapsing images and engulfment were included in a report on *Arabidopsis thaliana* after Cr(VI) exposure [43], further generalizing the view of a vacuolar-type PCD. 

Extensive protoplast condensation that results in the visible detachment of the plasma membrane from the cell wall was demonstrated in the cells of *P. sativum* under W stress with CLSM after FM4-64 staining [28]. In the present study, the same effect was confirmed by light microscopy (Figure 1E) and TEM (Figure 5A) in the same plant species after W exposure. The plasma membrane detachments create similar images with plasmolysis-associated vacuolar collapse in virus-induced hypersensitive cell death [42]. Cytoplasmic shrinkage manifested by plasma membrane detachment in epidermal cells was more generalized than in internal cells, denoting higher damage which is an observation further supported by the severe disruption of root epidermal cell files recorded in semithin sections (Figure 1C,E) and the complete degeneration of cytoplasmic organelles in epidermal rather than in cortex cells (Figure 5C,D). Epidermal cells appear to be more severely affected than the internal cortex cells since they are in closer contact to the stressing solution.This type of necrotic response might be considered as another kind of cell death [40]. However, the situation of a multicellular organ such as root exposed to an abiotic stress, currently W, might be quite different to that of suspension-cultured cells transferred to stressing growth media or submitted to heat shock [40,44] as all cells in the second case have an instant and direct contact with the stressful factor, and are highly vacuolated. Certainly, the types of PCD in plants remain a matter of fruitful discussion.

Our data show some differential effects in the degree and extent of W injury to the different cell compartments: vacuoles, ER and Golgi bodies seem to suffer the most severe structural damage, while mitochondria and plastids suffer the least as they keep some internal organization (Figure 2, Figure 3 and Figure 4). Organelle-specific malformations were previously reported for other heavy metals such as Cr (VI) in *Arabidopsis thaliana*, in which mitochondria and plastids suffered severe degeneration of their internal structure and became swollen and translucent, although they retained intact double-membrane envelopes [43]. In particular, the plastids assumed unusual and strange conformations. An explanation of these discrepancies is not evident, but they might be attributed to the differential distribution of the stressful factor within cell compartments, the difference in plant species and the differential induction of ROS by cell components in the presence of various abiotic stresses [32,45,46]. ROS, at a basal level, are considered to be essential for life supporting cellular proliferation, physiological function and viability, but they are thought to be the unavoidable byproducts of aerobic metabolism and, at higher concentrations, they become toxic for living cells [47]. In plants, ROS are routinely generated in chloroplasts, mitochondria, peroxisomes and other sites of the cell during normal metabolic processes such as photosynthesis and cellular respiration, their production being enhanced by abiotic stresses [33,48]. Many metals and metal nanoparticles, when in excess concentrations, induce ROS overproduction [43,49,50], a common feature of which is the dysfunction of the mitochondrial electron transport chain [51]. 

In contrast, evidence obtained in this study shows that W, under our experimental conditions, did not enhance the production of ROS, as indicated by three different experimental procedures including DCF-DA staining, NAC treatment, and the spectrophotometric analysis of the extracts after NAC exposure and Evans blue staining (Figure 7). If so, W-induced PCD might not be accomplished via ROS, which would render mitochondria and plastids more susceptible to metal toxicity rather than metal reaching and accumulating in these organelles [43,51]. This might provide an explanation why mitochondria and plastids were less degenerated than other organelles, an assumption that requires further documentation. 

It is well known that the ER undergoes dramatic changes in distribution and occupies strategic locations, influencing local events by taking up or releasing Ca^2+^ ions, during the cell division cycle [36,52]. Many processes, especially those based on the cytoskeleton, are very sensitive to Ca^2+^ changes. Therefore, the ER is closely involved in the normal progress of mitosis, during which the MTs undergo dynamic changes manifested through the organization of successive arrays correlated with specific mitotic phases, which may be disrupted by external factors [37,53]. Our control images for ER (Figure 6A–E) are in good accordance with the strategic role of ER in mitosis [52]. However, after W treatment, ER morphology and distribution were severely disturbed, confirmed by both TEM (Figure 3) and CLSM (Figure 6F–I), also reflecting W-caused MT disruption [24,28]. Moreover, the structural evidence obtained here indicates that the concentric conformations of ER recognized by TEM (Figure 3A,B) may be identical with the dense masses seen by CLSM after ER immunolocalization (Figure 6F,G). This conclusion is supported by the fact that both were observed only in W-treated cells and displayed circular conformation after both methodologies. The thick and concentric aggregations of ER in the cytoplasm and around the nucleus may be a common response to metal stress as they were also detected by TEM after Cr (VI) exposure in *Arabidopsis thaliana* [43].

Another role attributed to the ER in adverse conditions is its presumed involvement in the induction of a kind of PCD. Accumulating evidence indicates that many external abiotic and biotic stresses and developmental signals can induce ER stress, which in turn may be involved in specific types of PCD [54]. In addition, water stress has been shown to induce PCD via the ER stress pathway [55]. The structural malformations of W-treated ER in both the observed interphase and mitotic cells are in good agreement with evidence showing that the ER stress-unfolded protein pathway may be involved in W-induced PCD [28]. Details of the mechanism and how it is performed represent interesting future research directions. 

## 4. Conclusions

Data obtained in the present study support the view that W, a rare but toxic heavy metal, causes the differential ultrastructural malformation of cell components, which can be related to induced PCD in root tip cells of *P. sativum*. Five groups of evidence were determined: 1. Vacuolar collapse favors a destructive type of PCD. 2. Invaginations of cytoplasmic portions along with vesicles and organelles within vacuoles resemble microautophagy. 3. There is a difference in the toxicity intensity between the epidermal and cortex cells of the meristematic zone, which is attributed to the closer proximity of epidermal cells with the stressing solution. 4. ER disruption, which is correlated with MT disturbance that underlies the derangement of mitosis, may be involved in W-induced PCD by the ER stress-unfolded protein pathway. 5. Under our experimental conditions, the induction of PCD is not mediated by elevated H_2_O_2_ concentrations, as indicated by DCF-DA, NAC and Evans blue staining protocols. Overall, the data support the view of a W-induced vacuolar destructive PCD without ROS enhancement—a notion that is worthy of further investigation.

## 5. Materials and Methods

### 5.1. Material Preparation for Light and Electron Microscopy

Seeds of garden pea (*Pisum sativum* cv. Onmard; Fabaceae) were pre-germinated for four days in an incubation chamber at 20 °C. Thereafter, 90 seedlings (3 sets of 30) of uniform size and morphology were selected and transferred to three containers filled with ½ strength Hoagland solution: one was used as control and the others contained 200 mg/L or 500 mg/L W solution provided as sodium tungstate (Na_2_WO_4_) [23]. Seedlings were grown in an incubation room for up to eight days at 20 °C, with a photoperiod of 14 h light (100 μmol m^−2^ sec^−1^), with daily change of the solutions. Representative seedlings were comparatively photographed using a Nikon F401S camera with a Micro-Nikkor 60 mm f/2.8D lens. These preliminary experiments were directive in deciding the final experimental concentration—200 mg/L W for 12 and 24 h.

Root samples of all experiments (controls, 200 mg/L W for 12 and 24 h) were fixed and processed for light microscopy and TEM, as previously described [23,28]. In particular, root tips of about 2 mm in length were excised with a razor blade and fixed for 5 h in 2% paraformaldehyde (w/v) and 3% glutaraldehyde (v/v), in 0.05 M sodium cacodylate buffer, pH 7.0, at room temperature. Samples were then post-fixed in 2% osmium tetroxide (w/v), similarly buffered for 5 h, dehydrated in a graded acetone series, treated with propylene oxide, embedded in Durcupan ACM resin (Fluka Chemie AG, Buchs, Switzerland) and polymerized in an oven at 60 °C for 60 h. Semithin sections (0.5–2 μM), cut on a Reichert-Jung Ultracut E ultramicrotome using glass knives, were stained with 0.5% toluidine blue O in 0.5% borax solution, and examined and photographed with an inverse Nikon Eclipse TE 2000-S microscope. Ultrathin sections (70–90 nm) were cut serially to semithin ones with a diamond knife, collected on 100 mesh or one slot copper grids covered with a formvar supporting film, double-stained with 2% uranyl acetate and lead citrate, and finally observed and photographed using a JEOL JEM 1011 electron microscope equipped with a Gatan ES500W digital camera. Digital images were optimized for contrast and brightness with Adobe Photoshop CS6 software with only linear settings.

### 5.2. ER Immunolocalization 

Excised root tips of 4-d-old pea seedlings treated as above (control, and 200 mg/L W-treated for 24 h), were fixed for 60 min in 8% (w/v) paraformaldehyde in PEM buffer (50 mM PIPES, 5 mM EGTA, and 5 mM MgSO_4_·7H_2_O), pH 6.8 [28]. After washing with PEM, the cell walls were digested for 40 min in 2% cellulase (Onozuka R-10, Serva), 2% macerozyme-R10 (Serva) and 0.4% β-glucuronidase in PEM. Thereafter, the root tips were gently squashed onto polylysine-coated coverslips, left to dry, extracted with 5% DMSO + 5% Triton-X 100 for 1 h and incubated overnight at room temperature with the mouse 2E7 (Santa Cruz, CA, USA) monoclonal antibody, the generous gift of Dr. Hartmut Quader (Biocenter, University of Hamburg, Hamburg, Germany), that recognizes some specific structural proteins of ER bearing the HDEL epitope (the tetrapeptide sequence histidine, aspartic acid, glutamic acid, leucine), in a 1:40 dilution [56]. After washing with PEM, the cells were incubated with a 1:80 dilution of FITC-anti-mouse antibody in the same buffer for 3 h at room temperature, followed by 1 h at 37 °C. Nuclear DNA was also fluorescently stained with propidium iodide in PEM so as to judge the cell division stage from the chromatin state. The coverslips were finally mounted in an anti-fade solution and specimens were examined with a Nikon D-Eclipse C1 CLSM, with an optical sectioning step of 0.20 or 0.30 μM. An exciter at 488 nm and a barrier at 515/30 nm, and an exciter at 543 nm and a barrier at 570 nm, were used for ER and DNA, respectively. Image recording was conducted with the EZ-C1 3.20 software according to the manufacturer’s instructions. Selected photographs were processed (only in linear settings) with Adobe Photoshop CS6.

### 5.3. In Situ Detection of Hydrogen Peroxide with DCF-DA

To evaluate whether W can induce the production of ROS, fluorescent staining with 2,7-dichlorofluorescein diacetate (DCF-DA, Sigma), which specifically detects hydrogen peroxide (H_2_O_2_), was performed [57]. Roots of the untreated and W-treated seedlings were incubated in the dark with 25 μM DCF-DA for 30 min [27,43]. For the positive control, some roots were first treated with 10% H_2_O_2_ for 24 h and then incubated with DCF-DA. After triple washing with double-distilled water, the specimens were examined under the Nikon D-Eclipse C1 CLSM, with excitation and emission wavelengths at 488 and 530 nm, respectively.

### 5.4. Estimation of ROS with NAC and of Dead Cells with Evans Blue 

The presumed involvement of ROS in W toxicity was additionally studied with N-acetylcysteine (NAC), a strong scavenger of ROS [57,58,59]. This experiment was carried out because it is well documented that W disorganizes the MTs [23,24,25,28], most probably indirectly via a mechanism depending on the in vivo antagonism of W for the Mo-binding site of Cnx1 protein [24]. In this paper, we used the state of the MTs (normal or disorganized) as a marker of W toxicity [25]. 

Four sets of 4-d-old seedlings were processed as follows: (i) Transferred to water for 24 h. (ii) Treated with 1 mM aqueous solution of NAC for 24 h. (iii) Treated with 200 mg/L W for 24 h. (iv) Initially transferred to 1 mM NAC for 24 h and then exposed for another 24 h in 1 mM NAC + 200 mg/L W. 

Then, some root tips of each experiment were prepared for the immunofluorescent localization of the MTs applying the protocol described above for ER with the difference that these samples were incubated overnight with a rat anti-*a*-tubulin 1st antibody (YOL 1/34, Serotec) followed by a FITC-anti-rat 2nd antibody for 3 h [37]. The specimens were examined with the Nikon D-Eclipse C1 CLSM. 

Other root tips of all four treatments were checked for cell viability by performing the specific Evans blue staining that selectively detects dead tissues and cells [60], according to a previously described protocol [28]. In brief, 10 randomly selected roots were incubated in a 0.25% aqueous Evans blue solution for 15 min at room temperature and left in distilled water overnight. The apical 5-mm parts of the roots were excised, the dye was extracted in an aqueous solution of 50% ethanol/1% SDS for 1 h at 50 °C, and the optical density of the extracts was measured at 595 nm with a spectrophotometer (SmartSpec Plus, Biorad, Herts, UK). Data were presented as the mean ± standard error (SE) of three independent experiments and statistically analyzed with significance at *p* ≤ 0.05. 

## Figures and Tables

**Figure 1 plants-08-00062-f001:**
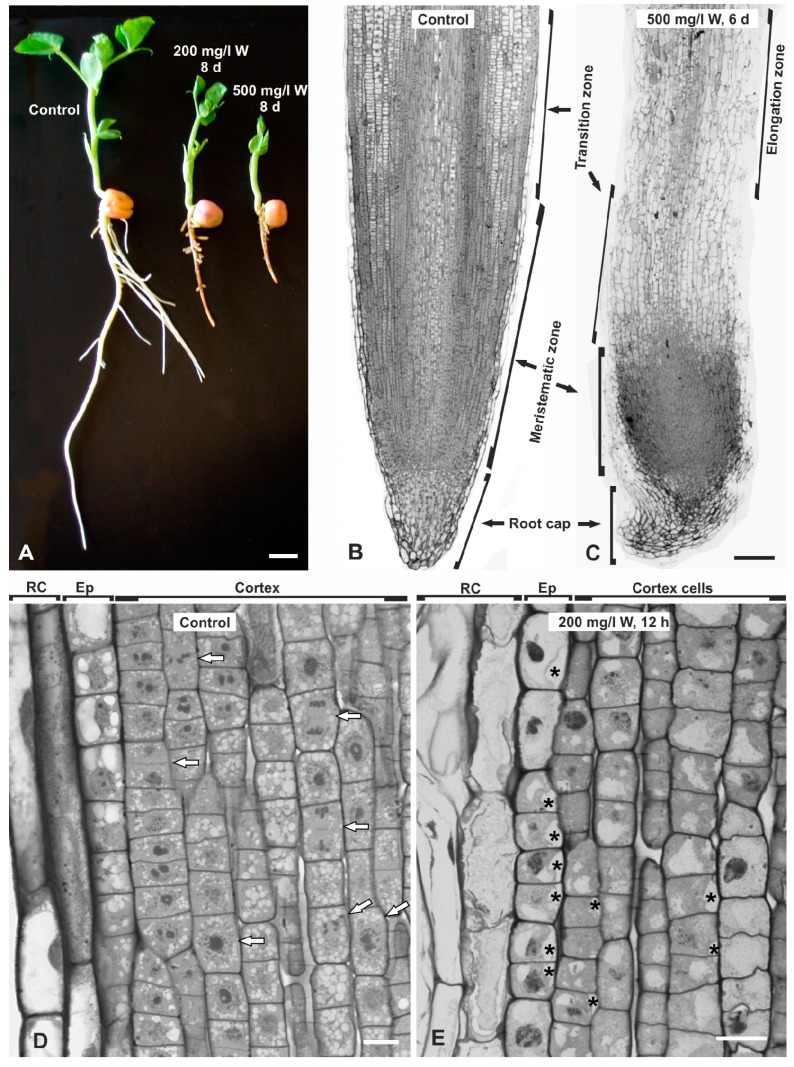
Phenotypic (**A**) and light microscopic (**B**–**E**) analysis of untreated and W-treated seedlings of *Pisum sativum*, as depicted. **A.** Control, 200 and 500 mg/L W-treated seedlings for 8 days. Note the concentration-dependent inhibition of seedling growth and the brownish color of the W-treated seedlings. **B**,**C**. Central longitudinal sections of untreated (**B**) and 500 mg/L W-treated roots for 6 days (**C**). W caused deformation of the root, shortening and crooking of the root cap, disruption of the vertical cell series and compaction of the meristematic zone towards the tip. **D**,**E**. Higher magnification views of the meristematic zone of untreated (**D**) and 200 mg/L W-treated roots for 12 h. Numerous cells at different mitotic phases are recognized in the control (**D**, arrows), but not in the W-treated roots (**E**). Asterisks mark detachments of the plasma membrane from the cell wall. RC = root cap cells, Ep = epidermis. Scale bars: A = 10 mm; B,C. 200 μM; D,E = 20 μM.

**Figure 2 plants-08-00062-f002:**
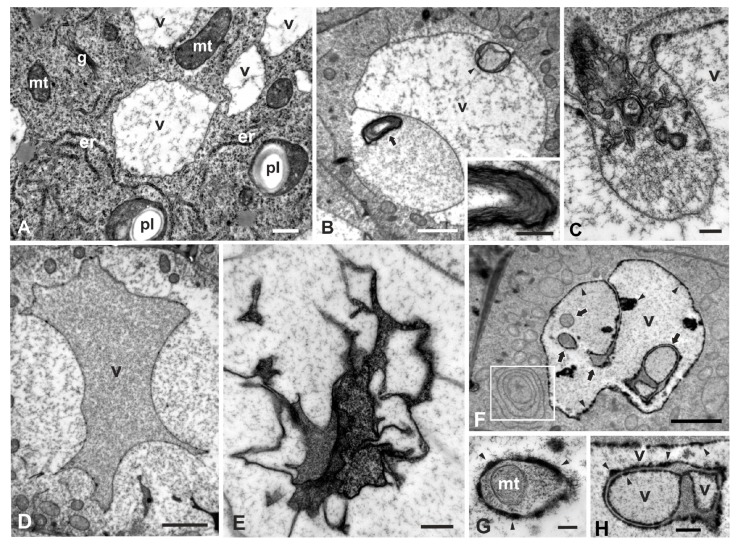
TEM micrographs of the effects of W on vacuoles of *P. sativum* root cortex cells. (**A**) Control. Small vacuoles (v) occur among endoplasmic reticulum (er), plastids (pl), mitochondria (mt) and Golgi bodies (g). (**B**–**D**) 200 mg/L W, 12 h. (**B**) A vacuole (v) containing a spherical membranous structure (arrowhead) and a large engulfment filled with granular material and a multilamellar body (arrow), part of which is magnified in the inset. (**C**) Engulfment of membranous, vesicular and amorphous material within a vacuole (v). (**D**) A collapsed vacuole (v) detached from the surrounding cytoplasm and filled with granular dense material. (**E**–**H**) 200 mg/L W, 24 h. (**E**) Accumulation of complicated membranous structures containing dense amorphous material within a vacuole. (**F**–**H**) Atypical vacuole (v) bearing transvacuolar cytoplasmic strands and protuberances (arrows), some of which are magnified in G and H showing the entrapment of a mitochondrion (**G**, mt) or complicated structures (**H**). All membranes are lined on the vacuolar face with electron-dense deposits (arrowheads). Outlined area is magnified in Figure 3B. Scale bars: A = 1 μM; B, D = 2 μM; B inset, E, H = 0.5 μM; C, F, G = 0.2 μM.

**Figure 3 plants-08-00062-f003:**
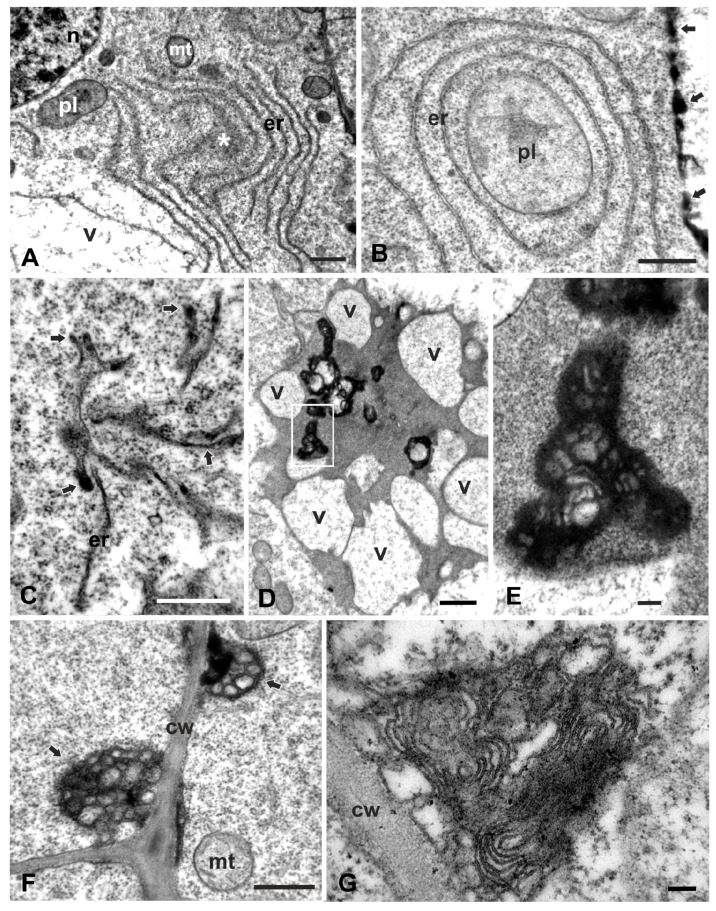
TEM micrographs illustrating the effects of 200 mg/L W for 12 h (**A**) or 24 h (**B**–**G**) on ER and on vesicular/lamellar conformations in cortex cells. Wavy bundle of endoplasmic reticulum (er) cisternae crossly or obliquely (asterisk) sectioned, close to a vacuole (v). pl = plastid, mt = mitochondrion, n = nucleus. (**B**). Concentric accumulation of ER (er) enclosing a plastid (pl) (enlargement of the outlined area in Figure 2). Arrows point to electron-dense material at the periphery of the nearby vacuole. **C**. Clustering of tubular elements with dense amorphous contents (arrows), contiguous with ER-like cisternae (er). (**D**, **E**). The central area of a cell containing several vacuoles (v) encircling amorphous material and electron-dense vesicular aggregations. Outlined area of Figure 3D is magnified in Figure 3E. (**F**, **G**). Multivesicular (**F**, arrows) and multilamellar (**G**) bodies in contact with the cell wall (cw). mt = mitochondrion. Scale bars: A, B, C, F = 0.5 μM; D = 1 μM; E, G = 0.1 μM.

**Figure 4 plants-08-00062-f004:**
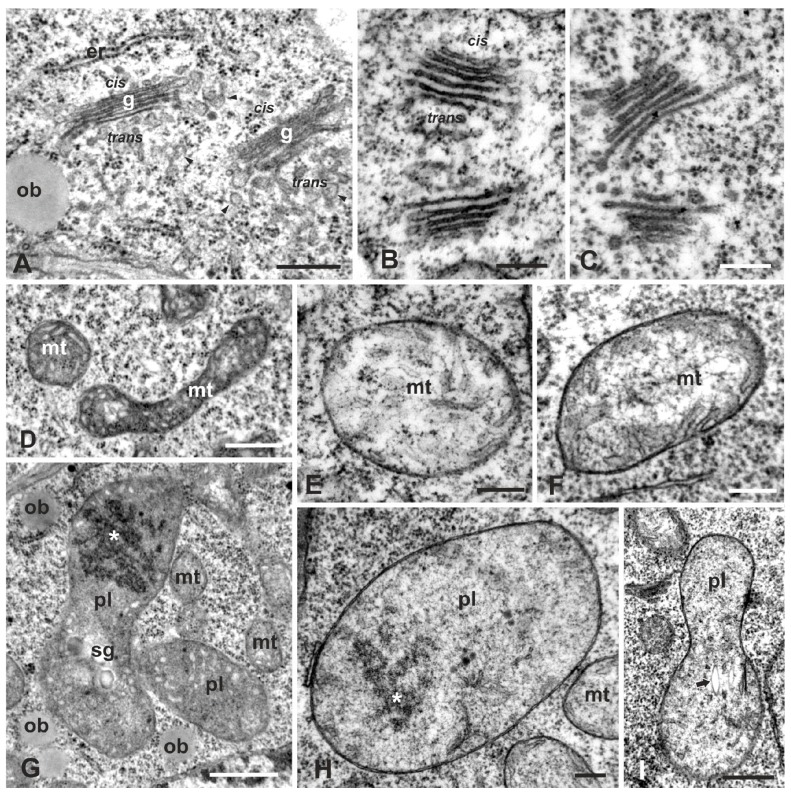
Effects of W on Golgi, mitochondria and plastids of cortex cells in the meristematic zone. (**A**). Control. Normal Golgi bodies (g) surrounded by numerous vesicles (arrowheads). ob = oil body. (**B**,**C**). Golgi bodies treated with 200 mg/L W for 12 h (**B**) and 24 h (**C**). In both treatments, Golgi bodies are similar in appearance, shorter than controls, highly contrasted and without any surrounding vesicles. **D**–**F**. Mitochondria (mt) of controls (**D**), 12-h (**E**) and 24-h (**F**) W-treated cortex cells. Control mitochondria display pleiomorphic shapes with profuse cristae (**D**), while in both W treatments they are oval and bear very few, barely viewable cristae in thin matrix (**E**,**F**). **G**–**I**. Plastids (pl) of control (**G**) and 24-h W-treated (**H**,**I**) cortex cells. Controls display varied morphology (**G**), while the W-treated ones are oval- or peanut-shaped, sparse stroma, without thylakoids or starch grains except for some remnants (arrow in I). Asterisks mark presumed protein deposits. ob = oil body, sg = starch grain. Scale bars: A, D, G, I = 0.5 μM; B, C, E, F, H = 0.2 μM.

**Figure 5 plants-08-00062-f005:**
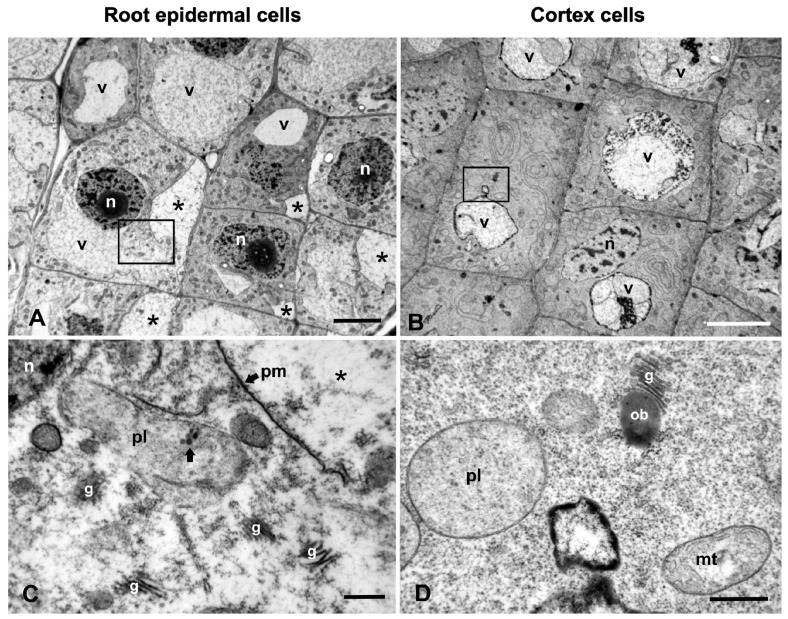
Comparative micrographs of epidermal and cortex cells in the meristematic zone of *P. sativum* roots treated with 200 mg/L W for 24 h. (**A**,**B**). General views showing many detachments of the plasma membrane in the epidermal cells (**A**, asterisks), but not in the cortex cells (**B**). Vacuoles (v) in both cases are atypical. (**C**,**D**). Higher magnification views of the outlined areas in (**A**) and (**B**), respectively, showing severe disintegration of plastids (pl) and Golgi bodies (g) in the epidermal cell (**C**), but less degraded in the cortex cell (**D**). n = nucleus, ob = oil body, pm = plasma membrane. Scale bars: A, B = 5 μM; C, D = 0.5 μM.

**Figure 6 plants-08-00062-f006:**
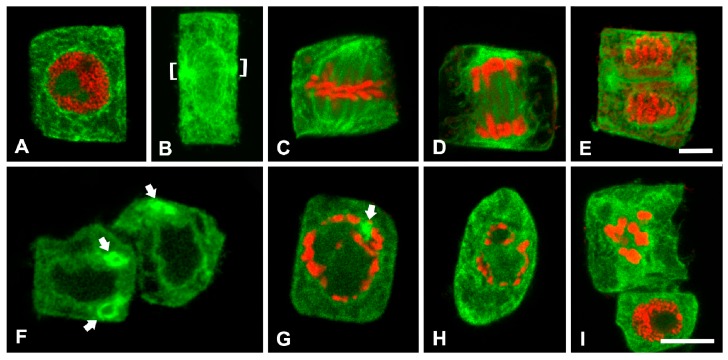
Confocal laser scanning microscopy (CLSM) images of ER (green) and DNA fluorescent staining (red) in *P. sativum* cortex cells of the meristematic zone. Single CLSM sections, except for B which is a projection. Upper row: control; lower row: 200 mg/L W-treated, 24 h. (**A**). Interphase cell. The ER is distributed evenly throughout the cytoplasm, while the nuclear envelope is also labeled. (**Β**). Preprophase cell with a preprophase band of ER (ER-PPB, brackets) and perinuclear labeling. (**C**–**E**). Metaphase (**C**), anaphase (**D**) and telophase (**E**) cells. The ER morphologically coincides with the expected MT arrays for the respective phases [37]. **F**. Interphase, W-treated cells. Atypical ER aggregations in bright, ring-like conformations (arrows). (**G**–**I**). Prophase W-treated cells. Atypical occurrence and distribution of ER, with some bright accumulations (arrow), and abnormal chromosome arrangement. Scale bars: 10 μM.

**Figure 7 plants-08-00062-f007:**
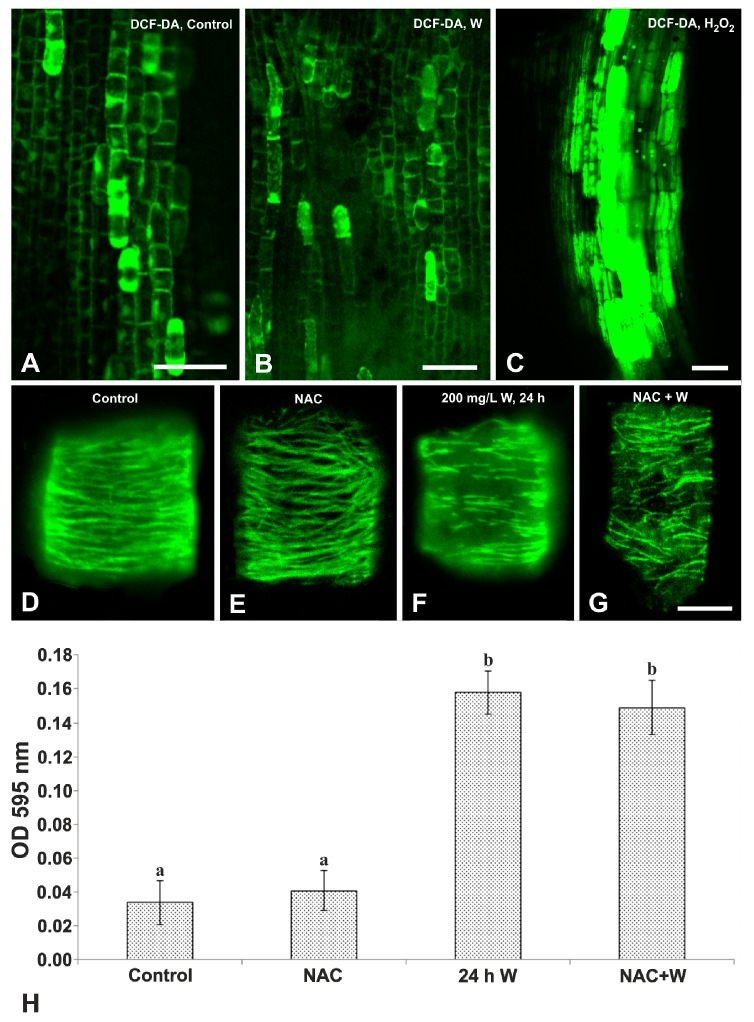
Determination of reactive oxygen species (ROS) production by DCF-DA staining and N-acetylcysteine (NAC) treatment. (**A**–**C**). Single CLSM paradermal sections in the meristematic zone. Untreated root (**A**), treated with 200 mg/L W, 24 h (**B**) and with H_2_O_2_, 24 h (**C**). Only the positive control seedling (**C**) displayed intense staining of numerous cells. **D**–**G**. Projections of CLSM sections. MT immunolocalization in interphase cells of untreated (**D**), NAC-treated (**E**) and W-treated seedlings (**F**) and seedlings treated with a combined treatment with NAC for 24 h and then with NAC+W for 24 h (**G**). The cortical MTs in untreated (**D**) and NAC-treated seedlings (**E**) exhibit a typical organization, while they appear disrupted in W-treated (**F**) and NAC+W-treated (**G**) seedlings. **H.** Diagram illustrating the optical density (OD) at 595 nm of pea root extracts after NAC+W combination treatments as depicted, Evans blue staining and extraction. The ODs of untreated and NAC-treated extracts are low and similar, while those of W-treated and NAC+W-treated extracts are much higher. Values correspond to means ± SE of three independent experiments. Different letters represent statistically significant differences at *p* ≤ 0.05. Scale bars: A–C = 100 μM; D–G = 10 μM.

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
