# Peer review of "Structural Evidence of Programmed Cell Death Induction by Tungsten in Root Tip Cells of *Pisum sativum"

_plants, 2019, doi:10.3390/plants8030062_

Reviewer 1 Report

The review of the manuscript entitled "Structural Evidence of Programmed Cell Death Induction by Tungsten in Root Tip Cells of Pisum sativum", submitted to Plants.

I have read the manuscript with interest. This is well designed study, providing an evidence for W-induced vacuolar destructive PCD without ROS enhancement. In my opinion methods used for this experiment are chosen adequately and the results are set-out logically. This work is well documented, original and novel study that merits publication. 

I have only small remark. Did Authors really not observe the autophagy-like structures: autophagic bodies or autophagosomes during TEM observation? That would confirm that PCD really occurs in that experiments. It seems that in some cases (see Fig. 2B) it is clear that it is autophagy pathway of PCD. Please track your other documentation to check if autophagic-like structure are present or not. If not, please discuss it. 

Author Response

Reviewer 1

The review of the manuscript entitled "Structural Evidence of Programmed Cell Death Induction by Tungsten in Root Tip Cells of Pisum sativum", submitted to Plants.

I have read the manuscript with interest. This is well designed study, providing an evidence for W-induced vacuolar destructive PCD without ROS enhancement. In my opinion methods used for this experiment are chosen adequately and the results are set-out logically. This work is well documented, original and novel study that merits publication. 

Response

Thank you very much for your encouraging comments.

I have only small remark. Did Authors really not observe the autophagy-like structures: autophagic bodies or autophagosomes during TEM observation? That would confirm that PCD really occurs in that experiments. It seems that in some cases (see Fig. 2B) it is clear that it is autophagy pathway of PCD. Please track your other documentation to check if autophagic-like structure are present or not. If not, please discuss it. 

Response

Yes, there are many images showing a vacuolar collapse (such as Figs. 2D, 2E, 5B, revised version) and many others displaying engulfment of cytoplasmic material within vacuoles (e.g. Figs. 2B, 2C, 2F), and many more in our TEM archives. Also, images such as 3D may represent engulfed material. We agree that all these indicate a microautophagic-type of PCD, but presumably this was not well emphasized in our original manuscript. Now this point is better discussed (see track changes in Discussion). Thank you.

Reviewer 2 Report

Dear Editor,
the article "Structural Evidence of Programmed Cell Death Induction by Tungsten in Root Tip Cells of Pisum sativum " about the induction of PCD by tungsten, The authors used transmission electron microscope and confocal.
Tungsten induced the collapse and deformation of vacuoles, degraded Golgi bodies, increased the incidence of multivesicular and multilamellar bodies, and caused the detachment of plasma membrane from the cell walls.
English is sufficient in my opinion.
The results are very interesting and even the confocal images are clear and informative.
TEM images are a bit less clear, may be for the pdf compression? In this case if possible better contrasted images would be useful.

The authors write that mitochndria and plastids are less affected but Fig. 4C would suggest that at least plastids are strongly affected by elevated concentrations of tungsten.
Fig. 2B, 3A and 3B show typical autophagic activity. Some improvement of the discussion may be done under this point of view. Similar images can be found in
Papini A, S Mosti, W G van Doorn (2014) Classical macroautophagy in Lobivia rauschii (Cactaceae) and possible plastidial autophagy in Tillandsia albida (Bromeliaceae) tapetum cells. Protoplasma 251(3): 719-725.

Author Response

Reviewer 2

Dear Editor,

the article "Structural Evidence of Programmed Cell Death Induction by Tungsten in Root Tip Cells of Pisum sativum " about the induction of PCD by tungsten, The authors used transmission electron microscope and confocal.

Tungsten induced the collapse and deformation of vacuoles, degraded Golgi bodies, increased the incidence of multivesicular and multilamellar bodies, and caused the detachment of plasma membrane from the cell walls.

English is sufficient in my opinion.

The results are very interesting and even the confocal images are clear and informative.

Response

Thank you for your encouraging comments.

TEM images are a bit less clear, may be for the pdf compression? In this case if possible better contrasted images would be useful.

Response

We apologize if TEM images were not so clear. Apparently, the pdf compression may have a negative effect since our original images appear to be clear. We have revised all plates and tried to improve clarity and quality, while some images were replaced with better ones. Fig. 4 was fully revised. We hope that now quality is satisfacrory. Thank you.

The authors write that mitochndria and plastids are less affected but Fig. 4C would suggest that at least plastids are strongly affected by elevated concentrations of tungsten.

Response

This comment was very helpful. Our study was focused on cortex cells of the meristematic zone and, secondarily, on root epidermal cells for comparison. Indeed, mitochondria, plastids, Golgi bodies and ER of epidermal cells seem to be more severely degenerated than in cortex cells, apparently because epidermis is in close vicinity with the stressing solution. This was not very clear in our original manuscript, but now we tried to elucidate it: in order to show this difference comparatively an additional plate (Fig. 5) was added, as it was also suggested by another reviewer.

Fig. 2B, 3A and 3B show typical autophagic activity. Some improvement of the discussion may be done under this point of view. Similar images can be found in Papini A, S Mosti, W G van Doorn (2014) Classical macroautophagy in Lobivia rauschii (Cactaceae) and possible plastidial autophagy in Tillandsia albida (Bromeliaceae) tapetum cells. Protoplasma 251(3): 719-725.

Response

Thank you for your suggestion. We are aware of the papers by Papini and colleagues and we already have cited the publication by van Doorn & Papini 2013 [41], in which the several types of PCD are reviewed. In the suggested paper by Papini et al. 2014 two different types of PCD in tapetum cells are described in two different plants: a macroautophagy in Lobivia rauschii, (Cactaceae) and a plastidial autophagy in Tillandsia albida (Bromeliaceae). Our data are rather compatible with the microautophagy process reviewed by van Doorn and Papini (2013), thus we mostly depended on this article and have improved relevant discussion.

Reviewer 3 Report

Programmed cell death is part of normal plant development or it can be induced by various external stimuli such as abiotic stress. It is known that tungsten induces cell death in root of P.sativum, which is characterized by protoplast shrinkage, specific PCD gene expression and dependency on 26S proteasome or caspase-like aktivites (Adamakis et al. 2011). In presented manuscript, the authors complement their previous observations by analyzing the ER ultrastructure because their previous results suggest an involvement of ER unfolded protein stress during  this process. They have also analyzed other aspects of W-induced PCD using confocal and transmission electron microscopy.

General comments:

1)      I think that the experiments are adequate and also the figures and micrographs are of a good quality. However, it is unclear to me, which cells in the meristem have been analyzed during different experiments. In some of them, the authors claim that cortex or epidermal cells were analyzed whereas in some others it was „meristematic root tip cells“ if I understood correctly. Could authors explain why?

Here are some examples: Vacuole morphology was studied in the root tip/meristematic cells whereas changes in ER were analyzed in cortex cells? Or was it always cortex cells in the meristematic region?

The same applies for ER and ROS experiments (line 236-248) – which cells have been studied? E.g. In Fig 6A and B, it seems that different cell types are compared, given that the control cells are much bigger compared to the treated ones?

I think that it will be interesting to discuss if there is differential response to W depending of initial cell fate. Making a point that different cell layers within the meristm respond differently could highly improve the whole manuscript and also the scientific significance.

2)      Eventhough the manuscript is easy to read, I would like to ask authors to pay a special attention to always explain why certain experiments has been done. Examples: line 250 –  could you explain what is NAC and why it has been used? lines 249-261 – could you explain why are you analyzing MTs organization? etc.

3)      I would also recommend to write a conclusion of your observations at the end of each paragraph. This usually helps the reader to understand what is the main observation and what does it mean. E.g. lines: 225, 242, 248, …

4)      Could you also explain, why do you use 200mg/l Na2WO4? Is this concentration similar to the W concentrations in the poluted soils?

Minor comments:

-          Fig. 1A – if authors claim that W-stress inhibits root growth, this should be quantified.

-          line 92 – for easier understanding the peripheral root cam and epidermis shoud be highlighted in the figure.

-          line 185 – control cells

-          Fig. 6 A-C – which regions of the root are compared? To me it seems that cells in A, B are inside the root whereas C is on the root surface.

Author Response

Reviewer 3

Programmed cell death is part of normal plant development or it can be induced by various external stimuli such as abiotic stress. It is known that tungsten induces cell death in root of P.sativum, which is characterized by protoplast shrinkage, specific PCD gene expression and dependency on 26S proteasome or caspase-like aktivites (Adamakis et al. 2011). In presented manuscript, the authors complement their previous observations by analyzing the ER ultrastructure because their previous results suggest an involvement of ER unfolded protein stress during  this process. They have also analyzed other aspects of W-induced PCD using confocal and transmission electron microscopy.

General comments:

1)      I think that the experiments are adequate and also the figures and micrographs are of a good quality. However, it is unclear to me, which cells in the meristem have been analyzed during different experiments. In some of them, the authors claim that cortex or epidermal cells were analyzed whereas in some others it was „meristematic root tip cells“ if I understood correctly. Could authors explain why?

Response

Thank you very much for all of your comments.

It is true that in our original manuscript this point was not very clear. We focused almost exclusively (except for Fig. 1, where general / introductive images are shown) on studying comparatively the effects of tungstate on epidermal and cortex cells of the meristematic zone. We selected this zone since cells are very active not only in dividing but also in growing as all of their organelles are very active. Here, any interference with normal processes is fatal and evident as compared with the maturing and mature tissues, which contain cells with huge vacuoles.

Here are some examples: Vacuole morphology was studied in the root tip/meristematic cells whereas changes in ER were analyzed in cortex cells? Or was it always cortex cells in the meristematic region?

Response

All observations were carried out in the meristematic zone, which is yet covered externally by the cup-like projection of root cap cells around the root tip, tapering upwards, as shown in light micrographs (Figs. 1B-E). No cells of other zones (transition, elongation, root cap) were used. To make things more clear, we now added one additional figure (Fig. 5, revised version). Also, in figure legends and in results we tried to make clear if we refer to epidermal or cortex cells, always of the meristematic zone.

The same applies for ER and ROS experiments (line 236-248) – which cells have been studied? E.g. In Fig 6A and B, it seems that different cell types are compared, given that the control cells are much bigger compared to the treated ones?

Response

For TEM, we first cut central longitudinal semi-thin sections, which were directive to make ultrathin sectioning accurately in the meristematic zone. This approach is already mentioned in Materials & Methods, section 5.1, 2nd paragraph. Moreover, meristematic cells and their topology (epidermal, cortex, central cylinder) are readily recognized at the electron microscope. Thus, we believe that both vacuole and ER morphology was studied in cortex cells of the meristematic zone (Figs. 2 & 3). Other organelles such as Golgi, mitochondria and plastids belonged also to the same cells (Fig. 4). Further, the additional figure (Fig. 5, revised version), provides a direct comparison between epidermal and cortex cells.

For CLSM (Fig. 6 & Fig. 7D-G, revised version), due to enzymatic treatments on the root tips used and squashing (see section 5.2), eventually the cells are spread over coverslips and they lose their topology, thus it is not possible to estimate anatomically if cells are epidermal, cortex or belonging to the central cylinder. However, since epidermal cells are by far outnumbered by the thousands of cortex and central cells, we believe they all are internal cells of the meristematic zone. Moreover, they are always compared with control samples processed with the same methodology.

Regarding Figs. 6A and B (now Figure 7A, B), we believe they are the same cell type. There is only a slight difference in size that results from a difference in magnification, as this is depicted by scale bars. By the way, scale bars were checked again for all figures and corrected if necessary.

1)      I think that it will be interesting to discuss if there is differential response to W depending of initial cell fate. Making a point that different cell layers within the meristm respond differently could highly improve the whole manuscript and also the scientific significance.

Response

A difference was only observed between root epidermal and cortex cells (Fig. 5, revised version), which is attributed to the fact that epidermal cells are closer to the stressing solution. Any other differential response to W toxicity depending on initial cell destination could not be established in this study.

2)      Eventhough the manuscript is easy to read, I would like to ask authors to pay a special attention to always explain why certain experiments has been done. Examples: line 250 –  could you explain what is NAC and why it has been used? lines 249-261 – could you explain why are you analyzing MTs organization? etc.

Response

Some explanations were already given in Materials and Methods (sections 5.3 & 5.4), and additional ones were now added in the same sections. The relevant references were cited and now they were adjusted accordingly.

3)      I would also recommend to write a conclusion of your observations at the end of each paragraph. This usually helps the reader to understand what is the main observation and what does it mean. E.g. lines: 225, 242, 248, …

Response

In many cases there was already a short conclusion of the main observations, which however was not always at the end of each paragraph; instead, it was at the end of a piece of information (now highlighted by track changes). Now we tried to add where missing, taking at the same time care to be short so as not to disrupt page formatting, which was difficult to manage due to figures that cannot be cut and allocated in two pages. We hope to meet your approval.

4)      Could you also explain, why do you use 200mg/l Na2WO4? Is this concentration similar to the W concentrations in the poluted soils?

Response

The concentration used and timing were determined after preliminary experiments described in the first paragraph of section 5.1, but also depending on our previous experience and international bibliography.

The concentration used is higher than that in the polluted soils except for some highly polluted areas (e.g. Pratas J., Prasad M.N.V., Freitas H., Conde L. 2005. Plants growing in abandoned mines of Portugal are useful for biogeochemical exploration of arsenic, antimony, tungsten and mine reclamation. Journal of Geochemical Exploration, 85, 99-107). Moreover, as the reviewer pointed, this work complements our previous observations (Adamakis et al., 2011) in which the same W concentrations have been applied.

Minor comments:

-          Fig. 1A – if authors claim that W-stress inhibits root growth, this should be quantified.

This quantification has been done in a previous publication (Adamakis et al., Env. Exp. Bot., 2008).

-          line 92 – for easier understanding the peripheral root cam and epidermis shoud be highlighted in the figure.

The tissue topology is abbreviated on top of Figs. 1D & 1E and explained in the legend.

-          line 185 – control cells

-          Fig. 6 A-C – which regions of the root are compared? To me it seems that cells in A, B are inside the root whereas C is on the root surface.

For DCF-DA experiments (Fig. 7A-C, revised version) paradermal CLSM sections were examined containing cortex cells in the centre and maybe epidermal ones at the margins (due to the cylindrical shape of the root). For this approach, negative (Fig. 7A) and positive (Fig. 7C) controls were used. As already stated, the slight difference in size that results from a difference in magnification, is clearly depicted by scale bars.

Thank you for all of your suggestions, which were very constructive when revising this manuscript.